# An Analysis of Sport-Specific Pain Symptoms through Inter-Individual Training Differences in CrossFit

**DOI:** 10.3390/sports9050068

**Published:** 2021-05-19

**Authors:** Maria A. Bernstorff, Norman Schumann, Nader Maai, Thomas A. Schildhauer, Matthias Königshausen

**Affiliations:** 1Medical Department of Ruhr University of Bochum, University Hospital Bergmannsheil Bochum, Bürkle de la Camp-Platz 1, 44789 Bochum, Germany; nader.maai@bergmannsheil.de (N.M.); thomas.schildhauer@bergmannsheil.de (T.A.S.); matthias.königshausen@gmail.com (M.K.); 2Department of Mathematics, Ruhr-University Bochum, Universitätsstraße 150, 44801 Bochum, Germany; norman.schumann@ruhr-universität-bochum.de

**Keywords:** CrossFit, chronic pain, sports-related pain symptoms, repetitive microtrauma

## Abstract

Background: CrossFit is one of the fastest growing “high-intensity functional training” methods in recent years. Due to the very demanding motion sequences and high loads, it was initially assumed that there was an extremely high risk of injury. However, studies have shown that injury rates are given between 0.74–3.3 per 1000 h of training, which is not higher than in other individual sports such as weightlifting. The purpose of the study was to estimate the type of pain symptoms that are directly related to CrossFit, to estimate the frequency of injuries that occur within a population of recreational CrossFit athletes, and, finally, to identify the factors influencing the frequency of pain during CrossFit training. Methods: A total of 414 active CrossFit athletes completed an online survey inclusive of 29 items focusing on individual physical characteristics and training behavior, as well as simultaneous or previously practiced sports. Results: There was a significantly higher proportion of knee pain in athletes who had previously or simultaneously played another sport (*p* = 0.014). The duration, intensity, or type of personal training plan developed, along with personal information such as age, gender, or BMI, had no significant influence on the pain data. We could not find any significant variance between the groups that we formed based on the differently stated one-repetition max (RMs). There were differences in athletes who stated that they did specific accessory exercises for small muscle groups. Above all, athletes performing exercises for the hamstrings and the gluteus medius indicated fewer pain symptoms for the sacro-iliac joint (SIJ)/iliac and lower back locations. Conclusions: It is important not to see CrossFit as a single type of sport. When treating a CrossFit athlete, care should be taken to address inter-individual differences. This underlines the significant differences of this study between the individual athletes with regard to the ability to master certain skills or their previous sporting experience. The mere fact of mastering certain exercises seems to lead to significantly more pain in certain regions. In addition, there seems to be a connection between the previous or simultaneous participation in other sports and the indication of pain in the knee region.

## 1. Introduction

CrossFit is one of the fastest growing “high-intensity functional training” methods in recent years. Until 1995 it was limited to military sports [1]. Since then, it spread across the world. Today there are over 15,000 CrossFit training facilities worldwide [1].

CrossFit is both a training method and a competitive sport. All exercises can be reduced to a “scaled” variant, so that athletes can train together at their individual level. Competitive athletes usually complete additional personal training with an employed coach to guide them through a performance-oriented training, in which a high proportion of classic weightlifting, gymnastics, and mobility training can be found [2].

With the advent of this new type of training, the implementation of mostly demanding, complex movement sequences and the moving of very high loads, a potentially high risk of injury was expected [3,4,5]. Several studies in recent years have shown that it is mostly shoulder, lumbar spine, or knee pain that often causes problems for athletes [2,4,5,6,7,8].

The currently known study situation showed very inhomogeneous results in many cases. In the studies published to date, injury rates between 0.74 and 3.3 per 1000 h of training were given. CrossFit ranks between injury rates similar to or slightly higher than in weightlifting and below those in soccer [9,10,11]. Some studies have even indicated that the indication of an injury frequency per training hour is insufficient for a sport like CrossFit [12,13].

There are some prospective studies that accompanied beginners in CrossFit training and recorded the injury rate in this population [8], as well as large-scale anonymous online surveys that asked about “injuries during CrossFit” in addition to personal data [6,10,12,14].

The purpose of the study was to estimate what kinds of pain symptoms as a possible precursor of injuries are directly related to CrossFit, how often those pain symptoms occur within a population of recreational CrossFit athletes, and which factors are influencing the frequency of pain while CrossFit training.

## 2. Materials and Methods

This study was a descriptive epidemiology data collection that was carried out by means of an online questionnaire answered by the athlete in the time from August 2020 to December 2020. A positive ethics vote according to the Declaration of Helsinki was given by the Ruhr University Bochum. A total of 414 athletes who actively participate in CrossFit training were included in total. Despite the COVID-19 pandemic, everyone was involved in active CrossFit training during the time of answering that questionnaire.

The survey was created to record CrossFit-specific pain symptoms in a large cohort. For this, the online tool “www.umfrageonline.de” (accessed on 25 February 2021)., a tool that can be purchased from the local University, was used. A total of 29 items was recorded in the survey (see Table 1). Personal data such as age, gender, weight, and height; training-specific data such as the period in which the respondent practiced CrossFit, the intensity of training, the type of training; and the classification of the intensity benchmark weights and mastered skills were determined. The focus was on examining the data in a differentiated manner. A crucial point of this questionnaire was the recording of simultaneous or previously practiced sports. With these items, an approach to the question of which pain symptoms are actually sport-specific and which are a possible later manifestation of previously or simultaneously performed sports was created. In advance, trainers and athletes were asked about common complaints during CrossFit, so that the questionnaire was specifically created to detect pain during training. In the area of CrossFit Sports, it can be assumed that the names for exercises (e.g., pull-ups, strict press, etc.) were known and described the same exercise for all. In order to avoid misunderstandings in the localization of pain, a picture was displayed showing the area. Similar to other works in this area, we refrained from a validation phase, because of a simple, clearly understandable data collection [5].

Empirically, it was shown that many CrossFit athletes previously did high-performance or very ambitious sports that have been shown to lead to physical limitations and pain in the long term. One aim was to find out which training-specific behavior had a significant influence on the pain reports of the participating athletes. For this, it was important to first find out whether the locations where pain was reported actually came from CrossFit-specific training, deferring to the prevailing opinion that sports that were previously or simultaneously practiced are hardly taken into account. This specific question was addressed by recording the types of sports the athletes played regularly before or during the time of the participation in CrossFit training. The question was whether athletes who had previously exercised regularly had pain significantly more often in certain locations than those who only participated in CrossFit. Each of the factors recorded in the questionnaire was examined on its influence on the frequency and region in which pain was reported.

In this questionnaire, all sports were recorded that were carried out simultaneously or before CrossFit training regularly. This information was identified in relation to the information of the pain region. Two groups were formed: a group without any previous sporting experience in addition to practicing CrossFit and a group with regular sport previously or simultaneously. The main focus of the current training consisted of CrossFit; other sports were either mainly carried out before CrossFit training or were only carried out in a temporally subordinate role.

The aim was to find out differences between the two groups in reports of pain in specific regions. In addition, both groups were categorized based on significant differences of age, gender, intensity, and duration of CrossFit training.

At one point, it was asked whether the athlete had ever had pain in connection with CrossFit and, if so, which joint/localization was affected (1 = hand/wrist, 2 = elbow, 3 = shoulder, 4 = cervical spine/neck, 5 = thoracic spine/upper Back, 6 = lumbar spine/lower back, 7 = SI joint/iliac, 8 = hip, 9 = knee, 10 = foot, 11 = ankle/ankle). This query was conducted in general and was not related to the performance of a specific exercise. For this purpose, the athletes were shown an illustration in order to avoid misunderstandings with the designation (see Figure 1). If there was a correlation to certain movements, the athlete then assigned the symptoms to one or more exercises in which the pain occurred most frequently in a free-text answer at the end of the questionnaire. 

The training-related data were primarily questions that related to sport-specific training around CrossFit. This information was to assess the intensity of the athletes’ exposure. The frequency per week and the years of doing CrossFit, the daily scope of training, and the performance in moving weights as well as mastered skills were queried. The specification of the RM (one-repetition max in kilograms) was not decisive at this point. It was in order to estimate at which level the respective athlete performed. In addition, it was asked how the athletes organized their training. Here, it was recorded whether a plan was used specifically designed for the individual by a trainer, a plan the individual designed on his or her own, or the program of the group training of a box as a guide. The question of accessory training for “rotator cuff”, “scapula retractor”, “hamstrings”, or “gluteus medius” was made with a view to possible injury prevention through this accessory work.

In addition, it was recorded whether and, if so, how regularly a warm-up training period and a cool-down were carried out, which could influence the probability of potential injuries. Another question regarding which factors influence the risk of injury and pain was the question regarding their participation in competitions. In the last question, the athletes expressed themselves in the form of a free-text answer for which exercise they felt pain during the execution. This was correlated to a corresponding localization (see Table 1).

### Statistical Evaluation

The statistics were created with Python 3.8 (Python Software Foundation. Python Language Reference, version 3.8, Scotts Valley, CA, USA) and Jupyter 1.0.0 (Jupyter Notebooks, Berkley, CA, USA). Packages used for the calculations and visualizations included pandas 1.2.0, numpy 1.19.4, seaborn 0.11.1, and scipy 1.5.4. As a statistical method, we used the Chi Square Test (Pearson or Fisher, depending on the sample size) throughout to test the stochastic independence in the contingency tables. The sample size of N = 414 was sufficient to make representative statements with the Chi Square Test. If there was a deviating size for partial observations, this has been indicated.

The statistical characteristics X and Y of the null hypothesis “H0: Characteristics X and Y are stochastically independent.” are described in the text. We always reported the *p*-value if the null hypothesis was significantly rejected for alpha 0.05. We did not consider other methods to be necessary, since, for example, a regression did not provide any added value for the question to be answered.

## 3. Results

Of the total of 414 participants there were 197 were men, 216 women, and 1 diverse, which corresponded to a distribution of men to women of 47% to 53%. The mean age in the entire group was 33.6 years, that of men 35.1 years and that of women 32.1 years. The BMI for all athletes was 25.1, for men 26.5 and women 23.9.

A total of 284 athletes reported having experienced CrossFit-related pain in the past six months. This corresponded to 72.1% of all respondents. Of the 284, 219 had to take a break due to pain, i.e., 55.6%.

In our data we found that the shoulder region (59.6% of all those who suffered pain and 37.4% of all athletes) was given as the most common pain location. The knee region followed at a considerable distance (35.4% with pain and 22.2% of all), closely followed by the lumbar region (31.9% with pain and 19.8% of all) (see Figure 2).

The athletes were asked about sports that they did currently and sports that they did previously in addition to CrossFit. This was recorded in a free-text answer. For easier data evaluation, all sports were categorized in assigned keywords (see Table 2).

### Frequency Distribution of Other Sports

Fitness sport was most often given together with other simultaneous or preceding sports. Overall, 252 participants responded “Yes” to the question of whether other sports were or are currently practiced in conjunction with CrossFit and 162 athletes responded “No” (see Table 2).

It was noticeable here that the indication of the pain localization “knee” was significantly higher in the group with previously performed sport than in the CrossFit group alone (*p* = 0,014). 

The training-related data were all set in relation to the individual pain regions.

There was no significant difference in any pain localization in relation to the length of CrossFit training in years. The frequency and intensity of the training did not show any significant difference in the individual regions with regard to pain symptoms either. Likewise, athletes who took part in competitions did not show a significantly higher number of pain reports in the individual regions.

A total of 91 athletes stated that they had their own training plan and that they trained accordingly. Most of these were created by a personal trainer. No significant difference was found in the frequency of pain comparing athletes with an exclusive personal training versus athletes participating in group training. 

For a more detailed examination of the so-called benchmark weights, the individual weights were categorized into five groups in order to be able to establish comparability (see Appendix A. At this point a total of 10 weight classes were created for each exercise asked. No significant difference could be found in any comparison.

The most striking results showed a correlation between a higher rate of reporting pain and mastering certain skills (see Appendix A, and Figure 3). It was not the question of whether the athlete experienced pain during the exercise, but rather correlated the ability to master a particular exercise with the fact that pain was reported in certain regions (Figure 3).

The table shows, on the one hand, the absolute values that reflect the number of athletes who had mastered a certain skill (e.g., strict pull-up) and, at the same time, the question of whether they had pain in a certain region (e.g., elbow). Since not all athletes were able to master a certain skill, the proportion of those who had pain in a certain location was sometimes greater than in the general population. For example, a total of 76 athletes were able to master the strict pull-up, 29 of whom also stated that they had pain in their elbows. This is a significantly high proportion (*p* = 0.02), even if no direct causality can be proven based on the question and the lack of investigation.

In the questionnaire, the four most common accessory exercises (exercises for the regions “rotator cuff”, “shoulder blade retractors”, “hamstrings”, and “gluteus medius”) were studied. In fact, in the summary of all localizations (see Appendix A) (upper body: hand, elbow, shoulder, cervical spine, and thoracic spine; lower body: lumbar spine, sacroiliac joint, hip, knee, ankle, and foot), the athletes doing accessory exercises indicated fewer pain symptoms for upper body as well as for lower body (“rotator cuff” vs. upper body: *p* = 0.007; “shoulder blade retractors” vs. upper body: *p* = 0.117; “hamstrings” vs. lower body: 0.001; “gluteus medius”: *p* = 0.003). A closer look at the individual regions shows this influence is most evident for “hamstrings” and “gluteus medius” exercises on the lumbar spine and SIJ (sacroiliac joint) (“hamstrings” vs. lumbar spine: *p* = 0.118; hamstrings vs. SIJ: *p* = 0.001; “gluteus medius” vs. lumbar spine: *p* = 0.001; “gluteus medius” vs. SIJ: *p* = 0.030).

The last question asked gave information on those exercises that led to pain in individual regions when performed. They are given in a crosstab as descriptive statistics (see Figure 4). Shoulder pain was particularly common during the execution of a “strict press” (N = 52), “pull-up” (N = 39), and “snatch” (N = 31); pain in the lumbar region during the “deadlift” (N = 42) and “backsquat” “(N = 37); hip pain with the “backsquat” (N = 35); and knee pain also with the “backsquat ”(N = 56).

## 4. Discussion

CrossFit is one of the fastest growing, high-intensity training methods in the world. Due to the very high loads and the often very complex movement sequences, it was assumed that the injury rate was high at the beginning of this sport era [5,7,8,9].

In the literature, the injury rate is often given as an injury rate per 1000 h of training. The injury rate in CrossFit ranges between 0.74 and 3.3 per 1000 h of training. [2,6,7,12,15]. On the one hand, this information is difficult to capture adequately due to a questionnaire that collects retrospective data, while on the other hand, we consider this to be insufficient for a sport like CrossFit. In CrossFit, the rate of acute injuries, as they occur in team sport or martial arts, is rather low. However, one can assume a high rate of pain symptoms and chronic impairments due to possible repetitive microtraumas. These should be the focus of further sports medicine considerations. There are already some studies that have started on this approach [16,17]. For example, a study from Oslo showed that the injury rates, as recorded to date, are far from reality. Using a questionnaire that they had athletes from various sports fields fill out every week, they determined the actual rate of so-called overuse manifestations. Their results often showed injury rates 10 times higher than the usual measurement methods [13]. Additionally, the number of injuries given in literature is inhomogeneous even for the number of acute injuries. “Injury” is not defined uniformly in the individual studies. In some works, the word “injury” is not defined at all for the athlete, so that some athletes only understand structural damage by it, while others also understand pain symptoms [6]. This should lead to a more differentiated way to detect sports-related injuries [13,17]. The aim of the present study was not to determine an injury rate, but to ask how many of the athletes had pain in certain locations of their body in the past six months. Often, pain can act as an indicator or harbinger of structural damage and should be viewed as a warning sign from the body that overuse or injury may already exist. The questionnaire should be viewed as a generous screening, even for athletes who have not yet consulted a doctor regarding their complaints and were, therefore, unable to give a diagnosis.

The data known to date show that CrossFit most often leads to injuries or pain in the shoulders, the lumbar region, and the knees. Depending on the study, the frequency of pain or injury differs between the lumbar region and the knee. However, the shoulder region is always the most mentioned region [6,7,8,9]. Our data revealed that the shoulder region (59.6% of all those who suffer pain and 37.4% of all athletes) was given as the most common pain location. The knee region followed at a considerable distance (35.4% with pain and 22.2% of all), closely followed by the lumbar region (31.9% with pain and 19.8% of all). The very high rates of pain determined here allow the hypothesis above that chronic damage to the musculoskeletal system could be more frequent than the injury rate prevailing in the literature suggests. However, further studies are necessary to determine the actual injury rate, as pain information cannot be equated with structural damage.

CrossFit is a sport with great inter-individual differences. On the one hand, this is due to the fact that CrossFit is made up of various sports such as gymnastics, weightlifting, and endurance sports [18,19]; on the other hand, the “typical” CrossFitter is often an ambitious athlete who had practiced other sports intensively in younger years or is still practicing other sports, as can be seen in Table 2. In our opinion, it is of great importance to take a closer look at the sports that were previously performed or that are still performed regularly. We wanted to supplement this data by examining the extent to which it can actually be said whether these are sport-specific pain symptoms or whether previous sports could have an influence on the pain reported by the athletes. There was a significantly higher proportion of knee pain in those athletes who had previously played another sport. As it can be seen in Table 2, these are often knee-intensive sports such as running, basketball, or soccer. Due to the fact of a retrospective data collection, no causality can be drawn but it can give an idea for further examination.

In addition, this study gave a closer look at the sport-specific differences in training behavior, as well as in performance-oriented aspects such as the implementation of some skills and personal RMs, and examined the extent that these influence the athlete’s pain information. As mentioned before, the sport CrossFit can only be viewed with difficulty as a uniform sport due to the influences from different sports such as gymnastics, weightlifting, and high-intensity training [18,19]. Our goal was to look at each athlete as individually as possible in order to find out which aspects can lead to the corresponding pain. Duration, intensity, the type of sport, how a personal training plan was created, or personal information such as age, gender, weight, height, or BMI had no significant influence on the pain information. There were differences in athletes who stated that they did specific accessory exercises for small muscle groups. Above all, athletes performing exercises for the hamstrings and the gluteus medius indicated fewer pain symptoms for the SIJ/iliac and lower back locations [20]. There was no significance between the groups that we formed based on the differently stated RMs.

The findings related to the RMs are difficult to interpret adequately due to the different physical constitution. The fact that, despite the higher loads, the pain indications did not increase significantly suggests that the technical execution in particular is decisive for the rate of pain indication. However, it must be admitted at this point that the technical performance of the mastered weightlifting movement could not be verified in this kind of study. This could also be investigated in further studies.

It was noticeable that the fact that they had mastered certain exercises showed a connection with the indication of pain. This connection between controlled exercise and significantly more pain in certain regions (see Figure 3) showed us most clearly how individually this sport must be viewed. It shows that it could only be certain movements in a certain sport that can lead to pain, which can be a precursor to structural damage.

In a retrospective data collection, as given, there are some limitations to discuss. With regard to the interpretation of the significant differences between the groups with and without sports experience in addition to CrossFit training, no causality can be drawn from this data, but it could encourage some consideration of sport-specific injuries and pain in CrossFit. Investigation of this in a prospective study may provide more information. Even if these data were collected retrospectively and no health status was carried out for the respective athletes before starting CrossFit training, a certain causality can be identified based on the relatively large group and the clearly significant difference to the group without simultaneous or previously performed sports suspected. It is possible that pain symptoms in the knee are often based on previous injuries and should not be seen as a purely sport-specific CrossFit injury.

## 5. Conclusions

We believe it is important to see CrossFit not as a single type of sport. When treating a CrossFit athlete, care should be taken to address inter-individual differences. This underlines the significant differences of this study between the individual athletes with regard to the ability to master certain skills or his/her previous sporting experience. The mere fact of mastering certain exercises seems to lead to significantly more pain in certain regions. In addition, there seems to be a connection between the previous or simultaneous participation in other sports and the indication of pain in the knee region. Athletes who regularly perform accessory exercises for the small muscle groups report less pain, especially in the lumbar region and in the sacroiliac joint.

## Figures and Tables

**Figure 1 sports-09-00068-f001:**
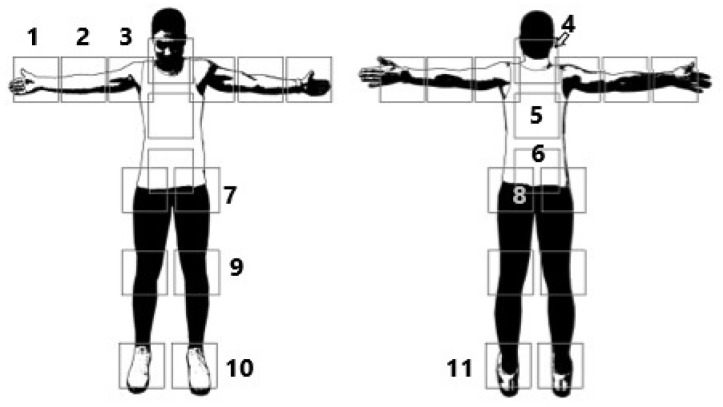
Schematic representation of the pain regions to be specified.

**Figure 2 sports-09-00068-f002:**
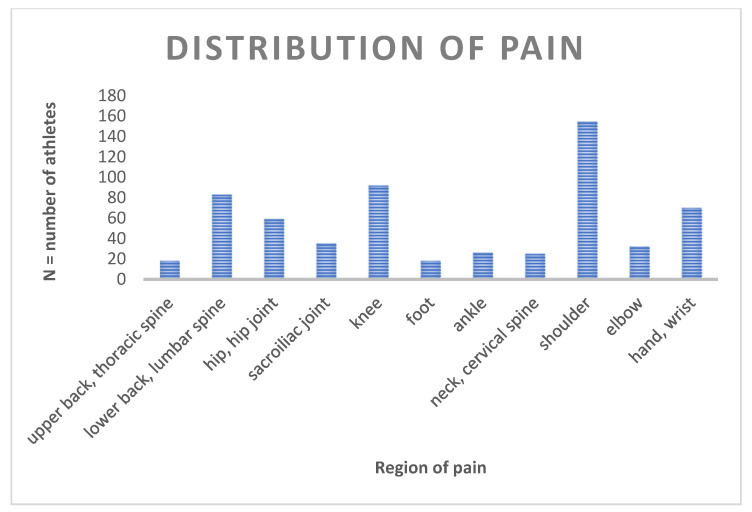
Distribution of the pain regions (N = total number of athletes mentioning a certain pain region; region of pain: regions with pain symptoms in connection with the CrossFit training).

**Figure 3 sports-09-00068-f003:**
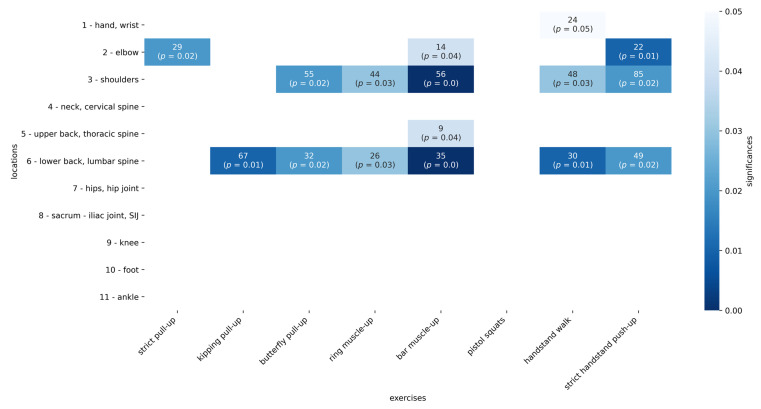
The upper number is the number of cases that both reported a pain symptom and mastered the skill. The resulting *p*-value is given in brackets.

**Figure 4 sports-09-00068-f004:**
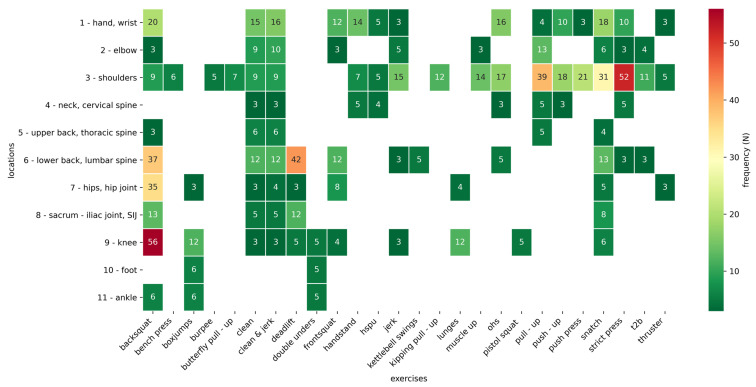
Overview of the most painful exercises: In this graphic, all exercises relating to pain localization are given in absolute value (abbreviations: hspu = handstand push-up; ohs = overhead squat; t2b = toes to bar).

**Table 1 sports-09-00068-t001:** Demographic profile of the CrossFit athletes.

Demographic Profile	
Sex N (%)	Male: 197 (47%) Female: 216 (53%) Divers: 1
Mean Age	Male: 35.1 years (21–58 years) Female: 32.1 years (18–50 years)
BMI	Male: 26.5 Female: 23.9
Weight	Male: 87 kg (64–140 kg) Female: 67.9 kg (45–105 kg)
High	Male: 182.4 cm (166–198 cm) Female: 168.4 cm (148–188 cm)

**Table 2 sports-09-00068-t002:** Frequency distribution of other sports: The list below shows the most frequently mentioned sports that were previously or were still performed besides CrossFit training. Individual responses were not listed (information both in absolute values and as a percentage of the total population (N = 414)).

Performing of Other Sports	
Fitness	133 (32.1%)
Running	70 (16.9%)
Soccer	49 (11.8%)
Martial Arts	48 (11.6%)
Swimming	36 (8.7%)
Handball	20 (4.8%)
Horsebackriding	19 (4.6%)
Racing bike	18 (4.3%)
Dancing	17 (4.1%)
Basketball	17 (4.1%)
Track and field	15 (4.6%)
Gymnastics	13 (3.1%)
Tennis	12 (2.9%)
Volleyball	12 (2.9%)
Mountainbike	11 (2.4%)
Triathlon	7 (1.7%)
Row	7 (1.7%)
Football	7 (1.7%)

## Data Availability

The data presented in this study are openly available in Bernstorff, M.A.; Schumann, N.; Maai, N.; Schildhauer T.A., Königshausen, M. An Analysis of Sport-Specific Pain Symptoms Through Inter-Individual Training Differences in CrossFit.

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
