# Peer review of "An Analysis of Sport-Specific Pain Symptoms through Inter-Individual Training Differences in CrossFit"

_sports, 2021, doi:10.3390/sports9050068_

Round 1

Reviewer 1 Report

This manuscript touches upon timely and important issue. However, I have some serious doubts as regards its publication.

Authors still confuse injuries and pain symptoms throughout the text. Injury is a physical trauma, damage caused by external force (accidents, falls, hits etc.). Study participant were asked about pain (question 12). Pain can accompany various and different situations or conditions (fatigue, DOMS, overtraining syndrome etc.). In the introduction and discussion, Authors refer to papers focusing on injury rates (per hours of training), thus this study in its current form is not fully relevant to the research area encompassing sport injuries. Discussing wit paper related to injury rate seems to be pointless. Pain and injury are two quite different areas (even though injury usually causes pain). Actually, the title should read e.g. "Distribution (occurrence?) of pain symptoms in recreational CrossFit participants". This study is about perceived pain (subjective), not actual injuries. Moreover, only joints but not muscles were included as the source of pain. 

Was the survey somehow validated for repeatability, precision or accuracy?

Many participants declared other sports. Was CrossFit their first/main sport or were they e.g. soccer players supplementing their training with CrossFit?

Statistical methods are not described. Indicating software package is not enough. 

The way of results presentation is very chaotic. There are data/table/figures that are unnecessary or could be moved to supplementary material. For me, the central information is that provided in fig. 4 where the distribution of pain symptoms as related to exercise mode and body area is shown. However, one row and one column should be added to show total numbers for exercise and body areas, respectively (fig. 2 is then redundant). Most 'painful' areas and exercise would be revealed. I totally cannot guess what is shown in fig. 3. Significances of what? Tables 4-6 are not necessary in the main text. What is the age range of the participants (minimum - maximum)? Standard deviations (if data normally distributed) or other measures of variability should be provided in table 2.

Having a quite big number of participants (not competitive athletes?), a more sophisticated method of data analysis could be used. The 'map' in fig. 4 is only an initial descriptive information. Author should go further and deeper and try to analyze correlations or relationships between pain symptoms and other available factors (age, sex, height, weight, BMI, performing other sports, absolute 1RM and as % of body mass etc.). A kind of multiple regression or factor analysis? What I find lacking in this paper is a reasonable design for analyzing the results and a clear convincing way of presentation.

Conclusions are not compatible with obtained results because of inferring about injury frequency/rate. structural damage, microtraumas etc.

Author Response

Review Report 1:

This manuscript touches upon timely and important issue. However, I have some serious doubts as regards its publication.

First of all, we want to thank you for the constructive comments made by the reviewer.

Authors still confuse injuries and pain symptoms throughout the text. Injury is a physical trauma, damage caused by external force (accidents, falls, hits etc.). Study participant were asked about pain (question 12). Pain can accompany various and different situations or conditions (fatigue, DOMS, overtraining syndrome etc.). In the introduction and discussion, Authors refer to papers focusing on injury rates (per hours of training), thus this study in its current form is not fully relevant to the research area encompassing sport injuries. Discussing wit paper related to injury rate seems to be pointless. Pain and injury are two quite different areas (even though injury usually causes pain). Actually, the title should read e.g. "Distribution (occurrence?) of pain symptoms in recreational CrossFit participants". This study is about perceived pain (subjective), not actual injuries. Moreover, only joints but not muscles were included as the source of pain. 

At no point it was intended to mix up injury rate and pain symptoms. The entire text has been revised in this regard in order to ensure that the results collected here only relate to the indication of pain and no manifest injury rate. The literature related to injury rate was included here because it can be assumed that pain can be a harbinger or an indicator of possible injuries. So if the reported pain is so much higher than the injury rate reported in the literature, there could be a discrepancy here, but that needs to be deepened in further studies. A final answer cannot be given from these data. The Paper is revised with the help of “Tracking changes” in Word, so you can see that this fact is clarified in the text.

Was the survey somehow validated for repeatability, precision or accuracy?

The questionnaire was collected primarily on the basis of empirical observations on the part of athletes as well as trainers and physicians. Validation tests were not carried out additionally before the survey.

Many participants declared other sports. Was CrossFit their first/main sport or were they e.g. soccer players supplementing their training with CrossFit?

This aspect has now been better elaborated and answered in the paper. These are athletes who mainly do CrossFit as their main sport and also do other sports or who did other sports more intensively before CrossFit.

Statistical methods are not described. Indicating software package is not enough. 

The statistical methods and the evaluation of the sample size is described in more detail in the paper.

The way of results presentation is very chaotic. There are data/table/figures that are unnecessary or could be moved to supplementary material. For me, the central information is that provided in fig. 4 where the distribution of pain symptoms as related to exercise mode and body area is shown. However, one row and one column should be added to show total numbers for exercise and body areas, respectively (fig. 2 is then redundant). Most 'painful' areas and exercise would be revealed. I totally cannot guess what is shown in fig. 3. Significances of what? Tables 4-6 are not necessary in the main text. What is the age range of the participants (minimum - maximum)? Standard deviations (if data normally distributed) or other measures of variability should be provided in table 2.

The presentation of the results has been adjusted so that a common thread should be easier to recognize. Graphics 3 and 4 have been revised. Graphic 3 was supplemented by the absolute values of those athletes who both mastered a questioned exercise and displayed pain in a certain region. We consider this representation to be a core element of this work, as we found significantly high numbers between those who master a skill and at the same time have pain in a certain region. A more detailed description should make this graphic easier to understand. Graphic 4 was made clearer in the sense that mentions up to two are no longer mentioned. The age range is now mentioned in table 2.

Having a quite big number of participants (not competitive athletes?), a more sophisticated method of data analysis could be used. The 'map' in fig. 4 is only an initial descriptive information. Author should go further and deeper and try to analyze correlations or relationships between pain symptoms and other available factors (age, sex, height, weight, BMI, performing other sports, absolute 1RM and as % of body mass etc.). A kind of multiple regression or factor analysis? What I find lacking in this paper is a reasonable design for analyzing the results and a clear convincing way of presentation.

In the method section, some graphics have been omitted to improve reading fluency. These are now attached in the appendix and mentioned in the running text with a reference to the graphic. In addition, the structure was adapted to show a more clearly visible central theme.

In the statistical analysis, all factors (age, gender, BMI, performing of other sports, RM values) were examined for their influence on the indication of pain. For the sake of clarity of the work, we have above all described the influencing factors with statistical significance in more detail and analyzed them in the discussion section. Many factors (gender, BMI, RM) showed no influence in our results. This is mentioned in the paper.

Conclusions are not compatible with obtained results because of inferring about injury frequency/rate. structural damage, microtraumas etc.

When presenting the results and when writing the discussion, care was taken to ensure that the results were pain symptoms and not injuries, so that the conclusion was drawn up compatible with the results obtained.

Reviewer 2 Report

First, I like to congratulate the authors for conducting such an interesting study. However,  the manuscript is not well prepared. Thus, the study leaves the reader with more questions than answers. There are many major concerns that the authors should address:
1) Citations are all over the place. There are only no. 17 and 18 in page 1. 
2) The sentences in the paragraphs are incomplete (suggest at least 4 sentences). 
3) There is no conceptual framework for the study?
4) The study was conducted during the COVID19 Pandemic, at least mention something on COVID and its relationship to the present study.
5) Table 1, is the translated questionnaire valid and reliable? Did the researchers test the translated version before it was used in the study?
6) Need to determine the power of your study. Is the sample size enough for your study?
7) Inferential statistics such as regression analysis should be conducted to determine the factor influencing the frequency of pain. 
8) Why fitness group have so much different in term of participation in this study? 
9) Does the study receive approval from the Human Ethics committee? Please specify. 
10. Discuss is very weak. Need rewriting and add limitation of the study. 

Author Response

Review Report 2:

First, I like to congratulate the authors for conducting such an interesting study. However, the manuscript is not well prepared. Thus, the study leaves the reader with more questions than answers.

First of all, we like to thank the reviewer for these constructive comments on this paper.

There are many major concerns that the authors should address:

1) Citations are all over the place. There are only no. 17 and 18 in page 1. 

We have revised the bibliography again, there may have been a formatting error, in any case all sources in our document were in place. There are a total of 21 sources in chronological order.

2) The sentences in the paragraphs are incomplete (suggest at least 4 sentences).

The text was checked again for grammatical errors by a native speaker.

3) There is no conceptual framework for the study?

CrossFit is not a new sport. It is made up of long-time practiced sports such as weightlifting and gymnastics. The combination of all these sports and the inter-individual differences between the athletes, however, harbors a new risk of injury. At this point, some data still have to be determined and evaluated. The present study is a screening for data determination of pain in CrossFit. The questionnaire is based on empirical observations and is intended to provide a basis for further more specific studies.

4) The study was conducted during the COVID19 Pandemic, at least mention something on COVID and its relationship to the present study.

This is a good objection. The aspect was taken up in the paper. From August to October the CrossFit boxes in Germany were open and all participants were able to do an active training.

5) Table 1, is the translated questionnaire valid and reliable? Did the researchers test the translated version before it was used in the study?

The German version was used to collect the data. A translation was only carried out for an English paper.

6) Need to determine the power of your study. Is the sample size enough for your study?

This was added in the paper. The sample size of n=414 is sufficient to make representative statements with the Chi Square Test. If there is a deviating size for partial observations, this has been indicated.  

7) Inferential statistics such as regression analysis should be conducted to determine the factor influencing the frequency of pain. 

The statistical survey was described and supplemented in more detail.

8) Why fitness group have so much different in term of participation in this study?

In the sports that are or have been carried out in addition to CrossFit training, the fitness group has the largest share. This can most likely be explained by the fact that many athletes have their first contact with dumbbell training in the gym and initially do not dare to practice such an elitist sport as CrossFit and only grow into it through fitness training. However, this is only a guess and is not the subject of this study and is therefore not included in the paper.

9) Does the study receive approval from the Human Ethics committee? Please specify. 

Yes, after the declaration of Helsinki, EthicsVotum was obtained by the Ethics Commission of the Ruhr University of Bochum.

10. Discuss is very weak. Need rewriting and add limitation of the study. 

The discussion was supplemented and limitations discussed.

Reviewer 3 Report

Abstract:

Change the term “high weight” to something more physiologically relevant such as heavy loads or high percentages of 1RM.

For the results aspect of the abstract this sentence is confusing if you do not read the entire paper “In relation to training behavior the data showed that the athletes doing accessory exercises indicated fewer pain symptoms (“hamstrings” vs. lumbar spine: p = 0.118; “hamstrings” vs. SIJ (sacroiliac joint): p = 0.001; “gluteus medius” vs. lumbar spine: p = 0.001; “gluteus medius” vs. SIJ: p = 0.030).” Could the authors potentially simplify this sentence for ease of a reader that more casually scans abstracts in this journal?

Introduction:

For this sentence “Competitive athletes usually complete additional personal training, in which a high proportion of classic weightlifting, gymnastics and mobility training can be found” what do the authors mean by personal training? Is it meant that they do other specific skill training or that they employ personal trainers to guide them through various activities?

Materials and Methods:

First, I’d like to commend the authors on an innovative way to assess pain on the body that circumvents the use of language that can be difficult for non-anatomists.

I think this sentence has a typo “but were by no means an exclusion criterion” but also, why was there no exclusion criteria? This could lead to problems of just detecting injuries in completely new exercises or those that have other risk factors that could lead to increased chances of musculoskeletal injury.

New section? The section that begins with “CrossFit is one of the fastest growing high intensity training methods in the world. Due to the very high weights and the often very complex movement sequences, it was assumed that the injury rate was high at the beginning of this sport era.[5,7–9]”.. perhaps this should be titled Discussion?

Conclusions and Other commentary:

I think that this is a very interesting paper. However, I feel it would be important to report and analyze more data. Please report their heights and weights (if I overlooked this I am sorry). Outside of that, I believe that the injury frequencies should be reported controlling for relative bodyweight strength as well as BMI. The idea of a “strength reserve” is a relatively well-known and supported factor regarding performance and risk of injury:

Suchomel, T. J., Nimphius, S., & Stone, M. H. (2016). The importance of muscular strength in athletic performance. Sports Medicine46(10), 1419-1449.

Outside of that oversight, I believe that this is a great paper and I would like to thank the authors for their work.

Author Response

Review Report 3:

Abstract:

Change the term “high weight” to something more physiologically relevant such as heavy loads or high percentages of 1RM.

The term is changed in the text to heavy loads.

For the results aspect of the abstract this sentence is confusing if you do not read the entire paper “In relation to training behavior the data showed that the athletes doing accessory exercises indicated fewer pain symptoms (“hamstrings” vs. lumbar spine: p = 0.118; “hamstrings” vs. SIJ (sacroiliac joint): p = 0.001; “gluteus medius” vs. lumbar spine: p = 0.001; “gluteus medius” vs. SIJ: p = 0.030).” Could the authors potentially simplify this sentence for ease of a reader that more casually scans abstracts in this journal?

The passage of the abstract has been simplified accordingly. This should improve the reading flow for an abstract.

Introduction:

For this sentence “Competitive athletes usually complete additional personal training, in which a high proportion of classic weightlifting, gymnastics and mobility training can be found” what do the authors mean by personal training? Is it meant that they do other specific skill training or that they employ personal trainers to guide them through various activities?

This is changed in the paper. These are personal trainers who are paid accordingly to create a training plan tailored to the individual strengths and weaknesses of the athlete.

Materials and Methods:

First, I’d like to commend the authors on an innovative way to assess pain on the body that circumvents the use of language that can be difficult for non-anatomists.

Thank you for this comment.

I think this sentence has a typo “but were by no means an exclusion criterion” but also, why was there no exclusion criteria? This could lead to problems of just detecting injuries in completely new exercises or those that have other risk factors that could lead to increased chances of musculoskeletal injury.

Exclusion criteria were primarily an age under 18 due to data protection regulations. Among the active CrossFit athletes, all athletes were under 60 years of age, so that a lower probability of chronic illnesses can be assumed. However, we wanted to do a general screening of all athletes and not a pre-selection.

New section? The section that begins with “CrossFit is one of the fastest growing high intensity training methods in the world. Due to the very high weights and the often very complex movement sequences, it was assumed that the injury rate was high at the beginning of this sport era.[5,7–9]”.. perhaps this should be titled Discussion?

Due to the importance of this statement, the content of this passage has been added in several places. This sometimes forms the basis for developing the idea for this paper. This was of course taken up in the discussion.

Conclusions and Other commentary:

I think that this is a very interesting paper. However, I feel it would be important to report and analyze more data. Please report their heights and weights (if I overlooked this I am sorry). Outside of that, I believe that the injury frequencies should be reported controlling for relative bodyweight strength as well as BMI. The idea of a “strength reserve” is a relatively well-known and supported factor regarding performance and risk of injury:

Suchomel, T. J., Nimphius, S., & Stone, M. H. (2016). The importance of muscular strength in athletic performance. Sports Medicine46(10), 1419-1449.

In the data analysis, all the data collected were analyzed for their influence on the target values. There were no significant differences in height, weight, BMI or personal RMs. As mentioned in the discussion, we consider the fact that the weight moved has no influence on the pain information to be difficult or impossible to discuss on the basis of this work. This would certainly be interesting to investigate whether it has something to do with technique, personal strength, or whether the physical constitution has an influence on it.

Outside of that oversight, I believe that this is a great paper and I would like to thank the authors for their work.

Thank you very much.

Round 2

Reviewer 1 Report

I really appreciate the work done by Authors but my opinion is still unfavorable, please forgive me my stubbornness. The manuscript was, admittedly, improved to some extent, however the most serious shortcomings of the study remain unchanged: the lack of a reliable/validated tool (questionnaire) and adequate statistical analysis. Besides, the Authors necessarily want to infer more from their data than is possible.  Below, my additional critical comments. I hope they will be helpful in future research attempts.

Title
"Frequency and entity of sport" - this part of the title is redundant and even incomprehensible. Especially the word 'entity', appearing just this once in the whole paper, is enigmatic.\

Abstract
How Authors "estimated the frequency of injuries" (as mentioned in the abstract), by which method? Where are the results of this estimation (e.g. predicted injury rate)?

Which factors influencing the frequency of pain during CrossFit training have been identified in the paper? By which method was the contribution of these factors calculated?

"The duration, intensity, or type of personal training plan developed, along with personal information such as age, gender or BMI, had no significant influence on the pain data." How was it calculated?

What is SIJ in the abstract? Not explained.

"It`s important not to see CrossFit as a single type of sport. When treating a CrossFit athlete, care should be taken to address inter-individual differences." Why such a conclusion? Earlier in the abstract one can read that most personal factors do not influence pain data (see above). Beside, these are speculations and postulates, not conclusions (that should be logically derived from the results).

Introduction
Some critical sentences:

Page 2 Line 46 "Several studies in recent years have shown that it is mostly shoulder, lumbar spine or knee pain that often cause problems for athletes. [2,4–8]" 

 Line 51 "In the studies published to date, injury rates between 0.74 and 3.3 per 1000 hours of training were given. CrossFit ranks between injury rates similar to or slightly higher than in weightlifting and below those in soccer. [9–11] Some studies have even indicated that the indication of an injury frequency per training hour is insufficient for a sport like Cross-54 Fit. [12,13]"

Line 56 "There are some prospective studies that accompanied beginners in CrossFit training and recorded the injury rate in this population [8], as well as large - scale anonymous online surveys that asked about "injuries during CrossFit" in addition to personal data.[6,10,12,14]"

Based on the above, what new adds this study to the existing knowledge? Did the Authors expect other results than reported previously? Why? Moreover, authors do not connect the literature data on injury rates with pain symptoms. Why Authors use "pain symptoms" instead of "injury rates" in their study? They are not the same but Authors treat them as if they were. The rationale and novelty of the study is not explained.

The study aims in the introduction (injury rates) are not fully consistent with those presented in the abstract (pain symptoms).

Material and methods
The questionnaire used in this study is not a validated tool.

In the method section, Authors write about study aims at some places. Shouldn't this be included in the introduction?

Chi Square Test - I do not think that, using solely this statistical tool, one is able to determine the pain factors and their contribution. Moreover, the main results were not controlled for several potentially confounding factors, such as age and sex in particular (why the Authors claim that these and other factors were of no importance?) The age range was were wide, from almost adolescents (18 years old) to the pre-retirement age. The effect of ageing (including potential vulnerability to injuries or injuries history) is expected to be present in such a diverse cohort. How was it controlled?

Results
Table 2 is Table 1, in fact. Why any measures of data distribution are not provided for age and BMI (standard deviation or interquartile range, depending on data distribution)? Also, since BMI is known, height and weight should be also known and shown.

What was the "frequency distribution of other sports" in age and sex groups (or divided by other factors)? Table 3 is, actually, Table 2 (the title should be above). The total percentage in this table is 124% meaning that more than 1 sport could be indicated and was presented. Were there any differences between multi- and single-sport participants (apart from CrossFit)? Besides, how were the percentages calculated, since the same 4.6% is equal to 19 participants at one place (horseback riding) and to 15 participants at another (track and field)?

Line 175 "pain localization “knee” was significantly higher in the group with previously performed sport than in the CrossFit group alone (p=0.014)" I think analogous statistical evidence should be provided for other factors (age, sex, training history, training frequency and intensity, competition, training plan, personal/group training etc.) to clearly demonstrate the differences or their lack (a table).

Line 192 "The most striking results showed a correlation between a higher rate of reporting pain and mastering certain skills (see table 6, and figure 3)." You mean a correlation coefficient? Was any used? In figure 3, one can see the  number of cases, not correlations. Significance of what statistical test is displayed on the right? By the way, table 6 does not exist...

Line 196 - Which 'table shows'? The explanation in this paragraph are very unclear/vague.

Page 8 Line 3 I cannot see Table 7 anywhere... You mean Fig. 4? Why not add a 'total' column and row as suggested in my previous review? Also, providing percentages (instead or next to absolute values) would give a more reliable picture of pain distribution, independent of the sample size. For example, 52 may be a high percentage in a small group, but virtually unnoticed in a very large sample of exercisers. Are 56 or 52 (of 414) cases really a high percentage (~13%) as suggested by the red color? Is there any critical 'limit' in one region that should be of concern?

Discussion
Page 10. The discussion starts here? The title of this section is not provided.
Line 21 "The aim of the present study was not to determine an injury rate..." This actually contradicts the statement in the abstract: "The purpose of the study was ....., to estimate the frequency of injuries...."

The discussion is focused on injury rates, although this was not the subject of the study. Of course, pain may (but not must) be an indicator of a potential injury but the causes of pain can be many and varied. But this study, due to its descriptive character, is not able to link pain symptoms and injuries. 

In general, the discussion just repeats information from the introduction and results sections, thus does not add further quality and interpretation to the obtained data.

Page 11 Line 59 "Duration, intensity, the type of sport, how a personal training plan was created, or personal information such as age, gender, weights, hights or BMI had no significant influence 60 on the pain information." How was it statistically demonstrated? (besides, 'weight' and 'height' are correct forms)

Lin 77 I think, the most serious limitations of this study are lack of reliable/validated tool (questionnaire) and adequate statistical analysis.

Conclusions
These are mostly speculations and postulates not directly based on study results ("We believe it's important to see...", "care should be taken to address...", "there seems to be a connection..." etc.). Conclusions must logically and directly follow from the study results and be consistent with study aims.

Author Response

I really appreciate the work done by Authors but my opinion is still unfavorable, please forgive me my stubbornness. The manuscript was, admittedly, improved to some extent, however the most serious shortcomings of the study remain unchanged: the lack of a reliable/validated tool (questionnaire) and adequate statistical analysis. Besides, the Authors necessarily want to infer more from their data than is possible.  Below, my additional critical comments. I hope they will be helpful in future research attempts.

Title
"Frequency and entity of sport" - this part of the title is redundant and even incomprehensible. Especially the word 'entity', appearing just this once in the whole paper, is enigmatic.\

This is a good point, we changed the title to: An analysis of sport-specific pain syndromes through inter-individual training differences in CrossFit

Abstract
How Authors "estimated the frequency of injuries" (as mentioned in the abstract), by which method? Where are the results of this estimation (e.g. predicted injury rate)?

At some points in the paper we emphasize that we did not determine an injury rate. Still, we believe that pain can be a common harbinger of injuries. There are certainly many CrossFit athletes who, although they are in pain, have not seen a doctor and therefore cannot report an injury. In order not to let them fall through the grid, we wanted to determine what pain each person is in.

Which factors influencing the frequency of pain during CrossFit training have been identified in the paper? By which method was the contribution of these factors calculated?

As can be seen from the results section, we were able to work out some factors that influence the pain symptoms of the athletes. The most important factors that produced significant differences were, above all, mastering of certain skills, previously or simultaneously performed sports and performing accessory exercises.

It was calculated by the Chi Square Test.

"The duration, intensity, or type of personal training plan developed, along with personal information such as age, gender or BMI, had no significant influence on the pain data." How was it calculated?

It was calculated by the Chi Square Test.

What is SIJ in the abstract? Not explained.

We changed it in the abstract. SIJ = Sacro – iliac joint

"It`s important not to see CrossFit as a single type of sport. When treating a CrossFit athlete, care should be taken to address inter-individual differences." Why such a conclusion? Earlier in the abstract one can read that most personal factors do not influence pain data (see above). Beside, these are speculations and postulates, not conclusions (that should be logically derived from the results).

As described in the results section, there are some factors that differ between individuals and show significant differences in the reporting of pain. The fact whether athletes had previously or simultaneously carried out a knee-stressing sport showed significant differences in the indication of knee pain, so that a cause from the CrossFit should not be assumed here. Whether an athlete has mastered certain skills showed clear differences in the reported pain, especially in the shoulders, elbows and lower back. In addition, athletes reported less pain when performing accessory exercises. For us, these are some inter-individual differences that justify the statement that CrossFit is not considered more uniform.

Introduction
Some critical sentences:

Page 2 Line 46 "Several studies in recent years have shown that it is mostly shoulder, lumbar spine or knee pain that often cause problems for athletes. [2,4–8]" 

 Line 51 "In the studies published to date, injury rates between 0.74 and 3.3 per 1000 hours of training were given. CrossFit ranks between injury rates similar to or slightly higher than in weightlifting and below those in soccer. [9–11] Some studies have even indicated that the indication of an injury frequency per training hour is insufficient for a sport like Cross-54 Fit. [12,13]"

Line 56 "There are some prospective studies that accompanied beginners in CrossFit training and recorded the injury rate in this population [8], as well as large - scale anonymous online surveys that asked about "injuries during CrossFit" in addition to personal data.[6,10,12,14]"

Based on the above, what new adds this study to the existing knowledge? Did the Authors expect other results than reported previously? Why? Moreover, authors do not connect the literature data on injury rates with pain symptoms. Why Authors use "pain symptoms" instead of "injury rates" in their study? They are not the same but Authors treat them as if they were. The rationale and novelty of the study is not explained.

In the studies cited, injuries were not uniformly defined. In such a large-scale survey, in our opinion, this is hardly possible due to the lack of an examination of the athlete. Anyone can reproduce pain themselves without a medical or sports-scientific examination. So we got a uniform statement. The novelty lies in the investigation of the inter-individual differences (skills, RMs, etc.). We wanted to define partial aspects of this sport, that have influence on the indication of pain. In addition, we had empirically noticed that many of the athletes who reported knee pain at CrossFit had very often done other knee-stressing sports before. This was part of our follow-up examination.

The study aims in the introduction (injury rates) are not fully consistent with those presented in the abstract (pain symptoms).

“The purpose of the study was to estimate, what kind of pain symptoms as a possible precusor of injuries are directly related to CrossFit, how often those pain symptomes occur within a population of recreational CrossFit athletes and which factors are influencing the frequency of pain while CrossFit training.“

Material and methods
The questionnaire used in this study is not a validated tool.

Legally empirical data should be recorded in the questionnaire presented here. These do not require a validation phase to be meaningful.

In the method section, Authors write about study aims at some places. Shouldn't this be included in the introduction?

“The purpose of the study was to estimate, what kind of pain symptoms as a possible precusor of injuries are directly related to CrossFit, how often those pain symptomes occur within a population of recreational CrossFit athletes and which factors are influencing the frequency of pain while CrossFit training.“

Chi Square Test - I do not think that, using solely this statistical tool, one is able to determine the pain factors and their contribution. Moreover, the main results were not controlled for several potentially confounding factors, such as age and sex in particular (why the Authors claim that these and other factors were of no importance?) The age range was were wide, from almost adolescents (18 years old) to the pre-retirement age. The effect of ageing (including potential vulnerability to injuries or injuries history) is expected to be present in such a diverse cohort. How was it controlled?

We examined every aspect (age, BMI, weight, height, gender) regarding the influence on the pain information. We found no significance in this population of 414 athletes.

Results
Table 2 is Table 1, in fact. Why any measures of data distribution are not provided for age and BMI (standard deviation or interquartile range, depending on data distribution)? Also, since BMI is known, height and weight should be also known and shown.

No, Table 2 is not table 1, it is just in the supplementary files, as wanted in the first revision. We added weight and high in table 2.

What was the "frequency distribution of other sports" in age and sex groups (or divided by other factors)? Table 3 is, actually, Table 2 (the title should be above). The total percentage in this table is 124% meaning that more than 1 sport could be indicated and was presented. Were there any differences between multi- and single-sport participants (apart from CrossFit)? Besides, how were the percentages calculated, since the same 4.6% is equal to 19 participants at one place (horseback riding) and to 15 participants at another (track and field)?

Table 3 is table 3 and table 2 is table 2! We did not conduct any further analysis between athletes who did one sport in addition to CrossFit and those who did two or more sports. The aim of this work was to identify an overview of factors that influence the athlete's pain report. In our view, further differentiation would make no sense in the context of this retrospective data analysis. However, this would be of interest for further studies in which one can examine the athletes and question them more precisely.

The percentage was calculated from the total population and rounded to one decimal place.

Line 175 "pain localization “knee” was significantly higher in the group with previously performed sport than in the CrossFit group alone (p=0.014)" I think analogous statistical evidence should be provided for other factors (age, sex, training history, training frequency and intensity, competition, training plan, personal/group training etc.) to clearly demonstrate the differences or their lack (a table).

We analyzed every factor that was assessed on the influence of the individual study populations. The work would lose its overview if every single aspect were discussed in the paper without statistical relevance.

Line 192 "The most striking results showed a correlation between a higher rate of reporting pain and mastering certain skills (see table 6, and figure 3)." You mean a correlation coefficient? Was any used? In figure 3, one can see the number of cases, not correlations. Significance of what statistical test is displayed on the right? By the way, table 6 does not exist...

Please note the supplementary files.

We only used the chi-square test because we wanted to find out the dependencies between the recorded factors.

Line 196 - Which 'table shows'? The explanation in this paragraph are very unclear/vague.

There is no table mentioned in Line 196 in the current version. It is not clear which passage is meant.

Page 8 Line 3 I cannot see Table 7 anywhere... You mean Fig. 4? Why not add a 'total' column and row as suggested in my previous review? Also, providing percentages (instead or next to absolute values) would give a more reliable picture of pain distribution, independent of the sample size. For example, 52 may be a high percentage in a small group, but virtually unnoticed in a very large sample of exercisers. Are 56 or 52 (of 414) cases really a high percentage (~13%) as suggested by the red color? Is there any critical 'limit' in one region that should be of concern?

Please note the supplementary files.

In the right column is the explanation for the coloring. Since this question is a free text answer, percentages cannot be given. For this we should have anticipated all possible answers so that one can determine the total population. The scope of the questions would go beyond the scope of such a questionnaire. This means that there is no suggestion of specifying a "total" column.

Discussion
Page 10. The discussion starts here? The title of this section is not provided.

Discussion is titled, page 11.  

Line 21 "The aim of the present study was not to determine an injury rate..." This actually contradicts the statement in the abstract: "The purpose of the study was ....., to estimate the frequency of injuries...."

In our view, this does not contradict what we said in the abstract. The abstract is only there for a summary and is intended to give an overview of the topic of the paper. The paper itself explains in more detail how we see the connection between pain and injury.

The discussion is focused on injury rates, although this was not the subject of the study. Of course, pain may (but not must) be an indicator of a potential injury but the causes of pain can be many and varied. But this study, due to its descriptive character, is not able to link pain symptoms and injuries. 

It was our goal that all athletes are recorded with pain, and not just those with diagnosed injuries. At no point do we presume that pain indication and injury are the same.

In general, the discussion just repeats information from the introduction and results sections, thus does not add further quality and interpretation to the obtained data.

Page 11 Line 59 "Duration, intensity, the type of sport, how a personal training plan was created, or personal information such as age, gender, weights, hights or BMI had no significant influence 60 on the pain information." How was it statistically demonstrated? (besides, 'weight' and 'height' are correct forms)

With the help of the Chi Square test, we examined the influence of each questioned item on the pain symptoms of the athletes.

Lin 77 I think, the most serious limitations of this study are lack of reliable/validated tool (questionnaire) and adequate statistical analysis.

Legally empirical data should be recorded in the questionnaire presented here. These do not require a validation phase to be meaningful.

Conclusions
These are mostly speculations and postulates not directly based on study results ("We believe it's important to see...", "care should be taken to address...", "there seems to be a connection..." etc.). Conclusions must logically and directly follow from the study results and be consistent with study aims.

A detailed examination of the subject cannot be carried out in a retrospective data evaluation with self-disclosure. It is intended to provide a basis for further studies.

Reviewer 2 Report

I had re-read the entire manuscript. It had been corrected as suggested, thus, I suggest accepting the manuscript. Thank you. 

Author Response

Thank you for your comments.